# Revealing the neural fingerprints of a missing hand

Sanne Kikkert[1,2†], James Kolasinski[1,3†], Saad Jbabdi[1], Irene Tracey[1,4], Christian F Beckmann[1,2,5], Heidi Johansen-Berg[1], Tamar R Makin[1*]

[1]FMRIB Centre, Nuffield Department of Clinical Neurosciences, University of Oxford, Oxford, United Kingdom; [2]Donders Institute for Brain, Cognition and Behaviour, Radboud University Nijmegen, Nijmegen, The Netherlands; [3]University College, Oxford, United Kingdom; [4]Nuffield Division of Anaesthetics, University of Oxford, Oxford, United Kingdom; [5]Department of Cognitive Neuroscience, Radboud University Medical Centre, Nijmegen, The Netherlands

**Abstract** The hand area of the primary somatosensory cortex contains detailed finger topography, thought to be shaped and maintained by daily life experience. Here we utilise phantom sensations and ultra high-field neuroimaging to uncover preserved, though latent, representation of amputees' missing hand. We show that representation of the missing hand's individual fingers persists in the primary somatosensory cortex even decades after arm amputation. By demonstrating stable topography despite amputation, our finding questions the extent to which continued sensory input is necessary to maintain organisation in sensory cortex, thereby reopening the question what happens to a cortical territory once its main input is lost. The discovery of persistent digit topography of amputees' missing hand could be exploited for the development of intuitive and fine-grained control of neuroprosthetics, requiring neural signals of individual digits.

*For correspondence: tamar. makin@ndcn.ox.ac.uk

†These authors contributed equally to this work

## Introduction

The hand area of the primary somatosensory cortex (S1) contains detailed digit maps, with physically adjacent digits represented next to each other. Using high-field neuroimaging, it is now possible to identify these characteristic digit maps in humans, with high inter- and intra-subject reliability (*Ejaz et al., 2015*; *Kolasinski et al., 2016*). Digit topography is characterised in neuroimaging by two main principles: digit selectivity (*Kolasinski et al., 2016*) and inter-digit overlap (*Ejaz et al., 2015*). These maps are thought to be shaped and maintained by daily life experience: digits used more frequently together in daily life benefit from increased representational overlap (*Ejaz et al., 2015*), and following single digit amputation remaining digits' topography changes (*Merzenich et al., 1984*).

Amputees commonly experience lingering sensations from their amputated body part (*Flor and Nikolajsen, 2006*). Phantom sensations are not necessarily painful, and are best described as a vivid sensation of the missing hand as if it is still present (*Flor and Nikolajsen, 2006*). When instructed to move their phantom hand, amputees report detailed kinaesthetic sensations regarding the extent of movement afforded by different phantom digits. Phantom movements are known to evoke signals in the sensorimotor system (*Makin et al., 2013a*; *Reilly et al., 2006*; *Raffin et al., 2012b*), previously attributed to abnormal processing caused by the amputation (e.g. aberrant inputs (*Makin et al., 2013a*), peripheral reorganisation (*Reilly et al., 2006*)). Here we interrogate the information content underlying activity elicited by phantom hand movements. If this information content is unchanged despite amputation, then activity patterns should show characteristic S1 digit topography.

**eLife digest** The brain has a remarkable ability to adapt to changes in circumstances. But what happens to the brain when it loses a key source of input, for example, following the amputation of a limb? A region of the brain known as primary somatosensory cortex processes sensory inputs from all over the body. The more sensitive an area of the body is, the more fine-grained its representation is in the cortex. For example, the hand is represented with a highly detailed map, with each finger represented seperately.

The brain is thought to require ongoing sensory signals from the body to maintain these detailed representations in the cortex. Indeed, textbooks typically state that the brain will 'overwrite' its representation of a body part if input from that area no longer arrives. According to this view, people who have lost a hand should show little or no activity in the area of primary somatosensory cortex that used to represent it.

However, many people who have had a limb amputated continue to experience vivid sensations of the missing limb long after its loss. When asked to move their so-called 'phantom' limb, these individuals report being able to feel the movement. Kikkert, Kolasinski et al. now show, using advanced imaging techniques, that the brains of individuals with phantom hands continue to represent the missing hand several decades after its loss. Indeed, asking the subjects to move individual fingers of their phantom hand activates fine-grained representations of those fingers, similar to those seen in two-handed controls.

By showing that the brain 'remembers' an amputated hand, Kikkert, Kolasinski et al. demonstrate that ongoing sensory input is not required to maintain representations of the body in somatosensory cortex. This, in turn, offers new hope for developing prosthetic limbs that are under direct brain control. If the brain continues to represent individual fingers many years after their loss, it should be possible to exploit those pathways to achieve intuitive fine-grained control of artificial fingers.

To study phantom digit topography, we used ultra high-field 7 tesla neuroimaging in two individuals who lost their left hand several decades ago (25 and 31 years post amputation) and eleven right-handed controls. Both amputees reported exceptionally vivid kinaesthetic phantom sensations during individual digit movements (see *Figure 1—source data 1*, *2*, for clinical and demographic details). Amputees and controls were visually cued to execute individual phantom digit movements (left hand digits in controls). Importantly, phantom movements are distinguishable from imagined movements. This is supported by empirical evidence demonstrating that phantom limb movements elicit both central and peripheral motor signals, that are different from those found during imagined movements (*Makin et al., 2013a*; *Reilly et al., 2006*; *Raffin et al., 2012b*, *2012a*). To ensure adequate task performance, amputees were asked to demonstrate to the experimenter outside the scanner the extent of volitional movement carried out in each of their phantom digits during the task, by mirroring the phantom movements with their intact hand.

To capture the first principle of topography, we employed a technique designed to identify digit preference in S1 (travelling wave design, see *Figure 1—figure supplement 1* for design details; see *Figure 1—figure supplement 2* for other brain areas activated by phantom hand movements). The resulting gradients of digit preference (hereafter digit maps) are presented in *Figure 1*. As indicated by the black arrows in the example control participants (*Figure 1A*), a characteristic digit map shows a gradient of digit preference, progressing from thumb (red, laterally) to little finger (pink, medially). Similarly, both amputees showed a clear gradient of digit preference in the central sulcus and post-central gyrus (*Figure 1B*; see *Figure 1—figure supplement 3* for intact hand maps). Qualitatively, the position, digit order, and extent of the missing hand maps were similar to those observed in controls. Analysis of spatial correspondence of 'same' versus 'different' digit clusters between two halves of the dataset further confirmed that while reduced (compared to controls), digit selectivity in amputees was consistent (*Figure 1D*; see Materials and methods for further details).

A hallmark of a functional sensorimotor system is the distinct representation of the two hands. To further confirm the existence of the missing hand map independently of intact hand contributions (*Makin et al., 2013b*), we designed an opposing bimanual travelling wave task. Participants

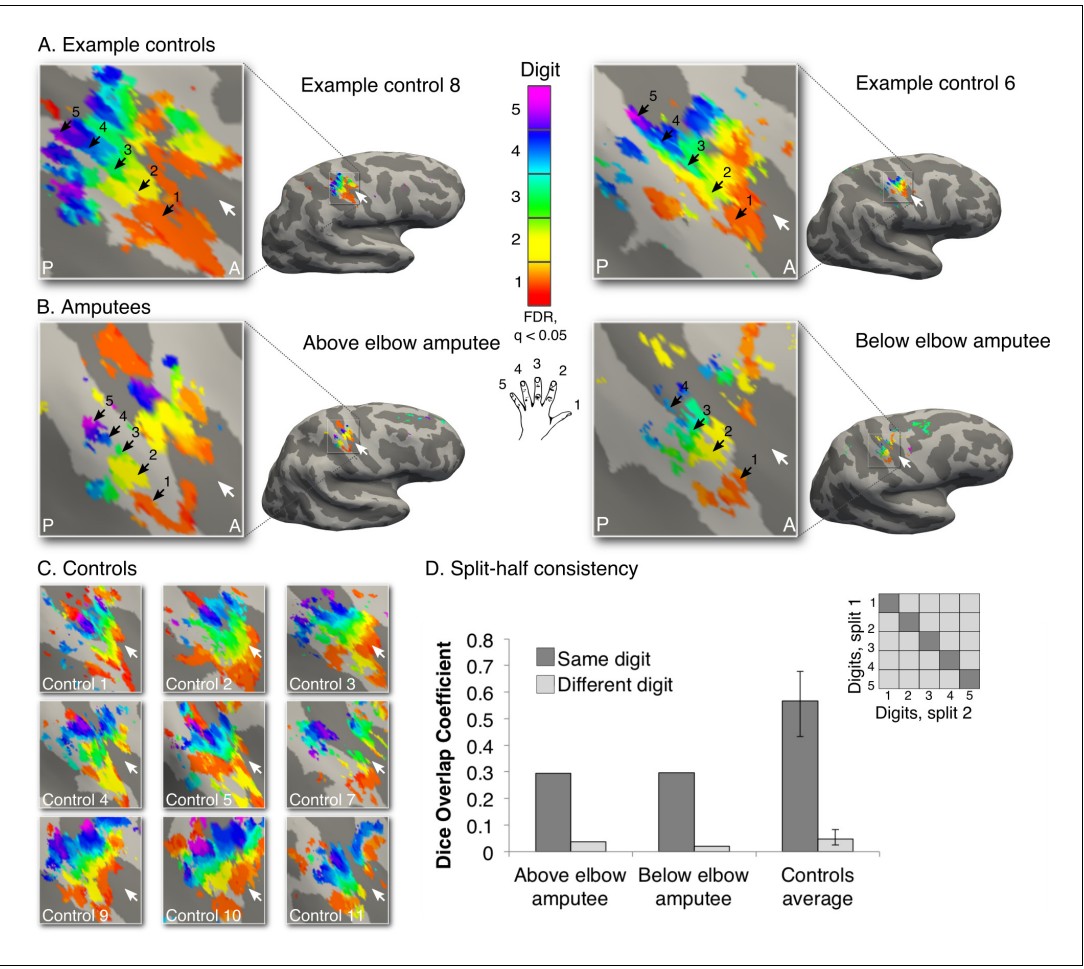

**Figure 1.** Missing hand maps revealed in amputees during phantom digits movement. Black arrows indicate preference for digits 1–5: thumb (red); index (yellow); middle (green); ring (blue) and little finger (purple) in two-handed controls (**A, C**) and amputees (**B**). Participants performed single digit flexion and extension movements with their non-dominant (controls) or phantom hand (amputees) in a travelling wave paradigm. Qualitatively similar digit topographies were found in each amputee and the controls. White arrows indicate the central sulcus. A = anterior; P = posterior. Multiple comparisons were adjusted using false discovery rate (FDR). (**D**) Maps' intra-individual split-half consistency, assessed using the Dice overlap coefficient. On average, 'same'-digit selective clusters (dark bars) showed greater consistency than 'different'-digit clusters (light bars) in amputees and controls. Amputees showed lower split-half consistency for 'same'-digit clusters (averaged across digits) compared to controls (95% confidence intervals (CI) = 0.43–0.68, as assessed using a bootstrap approach). However, amputees' 'same'-digit clusters were more consistent than 'different'-digit clusters in controls (i.e. fell outside the CI of 'different'-digit clusters split-half consistency in controls), indicating that although reduced, the digit maps of the amputees were consistent.

The following source data and figure supplements are available for figure 1:

**Source data 1.** Amputee demographic and clinical details.

**Source data 2.** Phantom digits movement vividness, difficulty and quality.

**Figure supplement 1.** Travelling wave task and analysis.

**Figure supplement 2.** Brain areas activated by phantom hand movement.

**Figure supplement 3.** Intact hand maps in amputees.

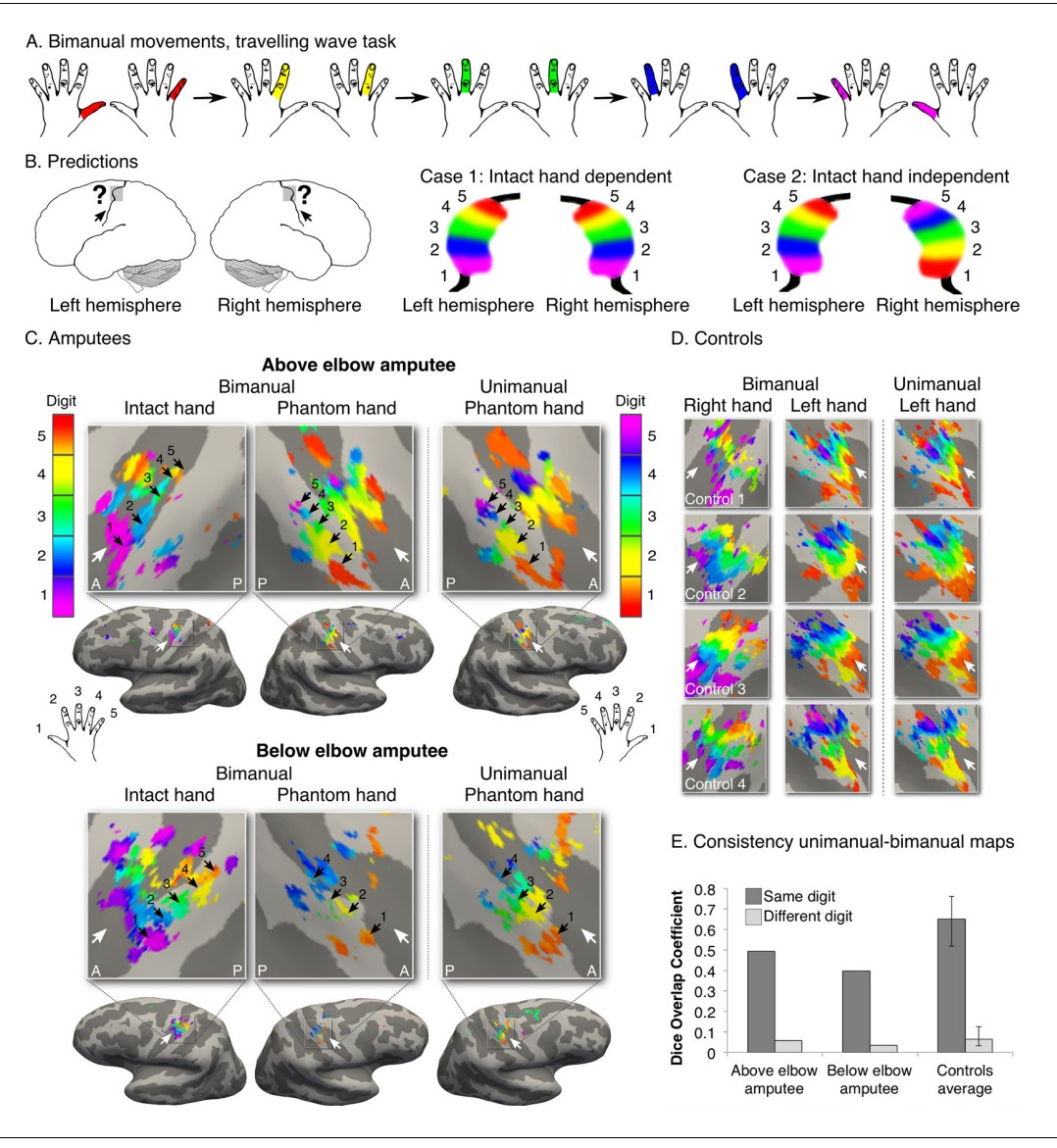

**Figure 2.** Missing hand maps are replicated during dissociated bimanual hand movements. (**A**) In the bimanual task, individuals performed paired digit movements with both hands, resulting in reversed digit cycles between the hands. The colour code indicates the cycle lag as shown in (**A**). Symmetrical colour gradients across the two hemispheres would reveal the missing hand map as dependent on intact hand movements (**B**: case 1). Colour gradient reversal across hemispheres indicates dissociated hand representations (**B**: case 2), as was seen in both amputees and controls (**C**, **D** respectively, left two panels). Maps contralateral to the missing hand (left hand in controls) resembled the unimanual task maps (*Figure 1*), both qualitatively (**C**, **D**, right panel) and quantitatively, using the Dice overlap coefficient (**E**). The digit maps were adjusted for multiple comparisons using false discovery rate (FDR). Other annotations are as in *Figure 1*.

The following figure supplement is available for figure 2:

**Figure supplement 1.** Over-representation of the intact hand in the missing hand territory.

performed paired movements with digits of the phantom and intact hands in reverse sequences (i.e. one hand moved from digit 1 to digit 5, while the other simultaneously moved from digit 5 to digit 1; *Figure 2A*). Both amputees showed maps in the hemisphere contralateral to the intact hand, dissociated from a second map in the missing hand hemisphere (*Figure 2B,C*). The latter map corresponded to the original (unimanual movements, *Figure 1*) missing hand map, as confirmed by

higher spatial correspondence for 'same' (average Dice overlap coefficient: 0.49, 0.40 for above and below elbow amputees) versus 'different' (0.06, 0.04) digit clusters across the two (unimanual and bimanual) missing hand datasets (*Figure 2E*). This provides a replication for the preserved missing hand map. This result also demonstrates the independence of the missing hand topography from the representation of the intact hand. Our group and others previously showed that following unilateral arm amputation, the intact hand becomes over-represented in the missing hand territory (*Makin et al., 2013b*; *Philip and Frey, 2014*; *Raffin et al., 2016*). Our results therefore confirm that the digit map in the missing hand territory is not driven by an emerging representation of the intact hand. Note, however, that our results do not exclude the possibility for reorganisation in the missing hand territory. As demonstrated in *Figure 2—figure supplement 1*, this study's participants showed over-representation of their intact hand in the missing hand territory, indicating that the preserved missing hand topography can co-occur with remapping of body parts.

While the travelling wave maps demonstrate digit preference, they provide little information about the second principle of digit topography: inter-digit overlap. To study inter-digit overlap, we used pairwise digit representational similarity of multivoxel patterns (*Diedrichsen et al., 2011*). This approach fully reveals the intricate overlap pattern across all digits (*Ejaz et al., 2015*), as shown for controls in *Figure 3B,D*. In general, both amputees showed greater average overlap across digits (0.55 and 0.54 for above and below elbow amputees respectively), compared to controls (95% confidence intervals (CI) = 0.24–0.41). However, when examining the inter-digit overlap pattern in the missing hand map, both amputees demonstrated typical patterns (*Figure 3A,C*; see *Figure 3—figure supplement 1* for intact hand). This was reflected in high correlation between each amputee's inter-digit overlap pattern and the controls, as assessed using a bootstrapping approach (average $r_s$ = 0.61 and $r_s$ = 0.78, 95% CI = 0.25–0.89 and 95% CI = 0.62–0.90 for above and below elbow amputees respectively). These average amputee-to-controls Spearman correlation values fell well within the controls-to-controls Spearman correlation range (95% CI = 0.35–0.95; *Figure 3C*), providing further evidence for characteristic missing hand representation decades after amputation.

Finally, we also studied missing hand representation in a third amputee (31 years since amputation) whose cause of amputation involved a brachial plexus avulsion, abolishing communication between the residual arm and the central nervous system. Due to MRI safety limitations 3 tesla neuroimaging was used (see *Figure 4B* for quality comparisons to ultra high-field digit maps). Using the bimanual task specified above (*Figure 4A*) we identified digit preference for the missing hand in the postcentral gyrus and central sulcus (*Figure 4C*). We also identified a typical inter-digit overlap pattern (*Figure 4D*), as confirmed in comparison with the control population described in *Figure 3* (average Spearman correlation with controls = 0.76, with 95% CI = 0.30–0.92). This result provides further evidence for the existence of preserved digit topography in the absence of peripheral input.

Together we show that, although the missing hand maps were weaker and noisier than the maps found in controls, the functional digit layout of S1 prevails following arm amputation. Digit topography, previously thought to depend on experience (*Ejaz et al., 2015*; *Merzenich et al., 1984*), was detectable despite decades without organised peripheral inputs associated with normal hand function. Our findings call for a reassessment of the role of sensory input in regulating brain organisation and plasticity. By demonstrating characteristic topography of the missing hand decades after arm amputation, our findings reopen the question of what happens to a cortical territory once its main inputs are removed.

Textbooks teach us that the cortical territory previously assigned to processing the now lost input is invaded by new representations. For example, following arm amputation, the missing hand territory in S1 is taken over by representations of other body parts (e.g. the neighbouring representation of the face in monkeys [*Pons et al., 1991*], or the intact hand in humans [*Makin et al., 2013b*]). Conversely, recent research in the visual cortex suggests that reorganisation in the adult brain may be restricted. For example following macular degeneration, the functional representation of the intact visual field was unchanged ([*Baseler et al., 2011*], see [*Smirnakis et al., 2005*] for similar results in non-human primates). Common to all these previous studies aiming to characterise reorganisation or the lack thereof, is that the authors probed the cortical neighbours of the area previously responsible for processing the lost input. While this approach is suitable for documenting shifted representation of the cortical neighbours, it leaves unexplored the possibility that the original function of the region deprived of sensory input may be preserved, though latent. Amputees experiencing phantom sensations provide a unique model to study what happens to the deprived

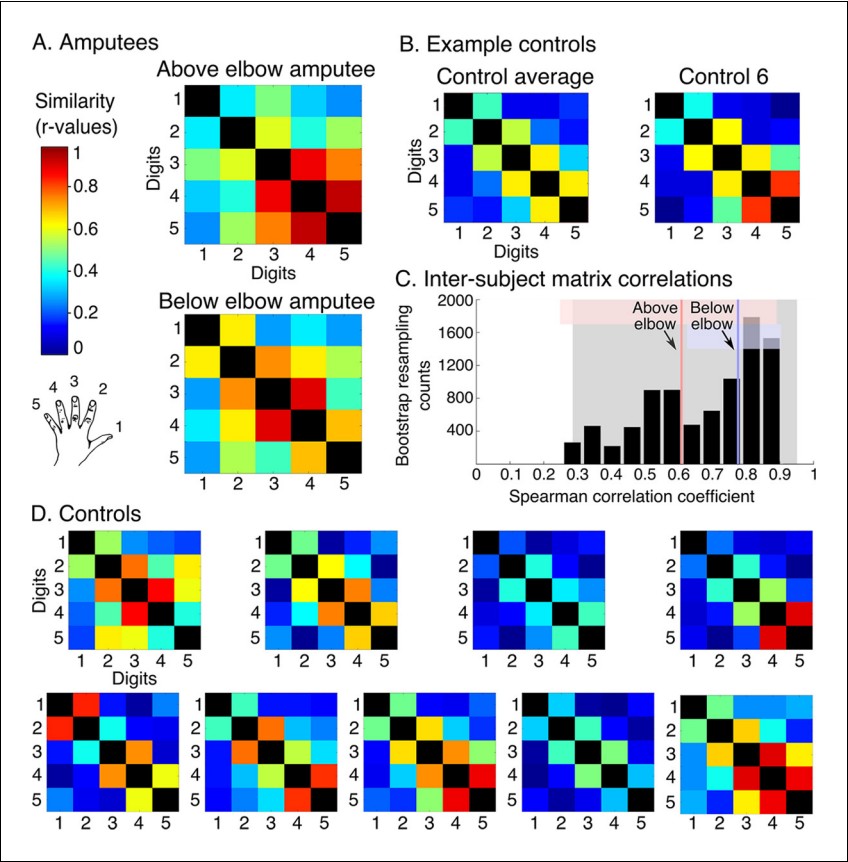

**Figure 3.** Amputees show characteristic inter-digit overlap patterns during phantom digit movements. Inter-digit representational similarity of multivoxel patterns underlying the maps shown in *Figure 1*, derived from a block-design paradigm in amputees (**A**) and controls (**B, D**). Similarity (or overlap) is decreased between non-neighbouring digits and tends to increase between digits 3–5, as shown in the controls' averaged matrix. (**C**) Positive distribution of inter-subject correlations, between controls' inter-digit overlap patterns. The grey area indicates the 95% confidence interval (CI) for controls. Pink and blue lines indicate average amputee-to-controls correlations for above and below elbow amputees respectively. Pink and blue shaded areas indicate the corresponding 95% CIs. Both amputee-to-controls correlation averages fell within the normal controls-to-controls correlation range, suggesting that the amputees exhibited a characteristic pattern of inter-digit overlap.

The following figure supplement is available for figure 3:

**Figure supplement 1.** Intact hand inter-digit overlap in amputees.

cortical territory itself during sensory input loss. Here we show that reorganisation in the missing hand territory following input loss does not abolish the original functional layout in sensory cortex.

How can our finding of preserved S1 topography of a missing hand be allied with the wealth of evidence showing cortical reorganisation in S1 following sensory input loss? (e.g. amputation and spinal cord injury [*Merzenich et al., 1984*; *Pons et al., 1991*; *Jain et al., 2008*]; see *Figure 2—figure supplement 1* for reorganisation in the current study's volunteers cohort). Already in their seal work Merzenich et al. suggested that reorganisation following sensory input loss does not exclude the possibility for simultaneous preservation of the original function of that region (*Merzenich et al., 1984*). Accordingly, recent structural and functional evidence shows that the capacity for S1 reorganisation is more limited than initially thought, and that instead the functional changes previously observed in S1 following input loss could be attributed to reorganisation in sub-cortical areas in the afferent pathway, principally the brainstem (*Jain et al., 1998*; *Kambi et al., 2014*; *Chand and Jain, 2015*). In other words, previous findings of massive cortical reorganisation in S1 reflect

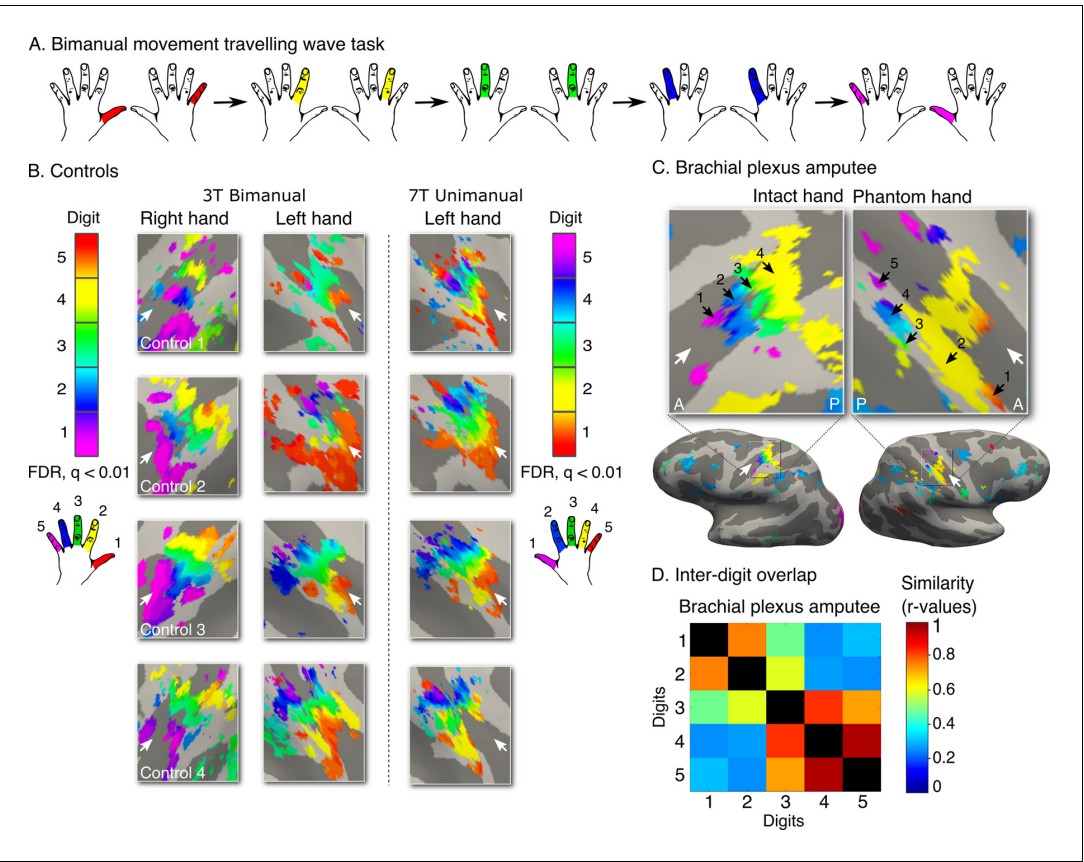

**Figure 4.** Topography is preserved despite the absence of peripheral input. To rule out the potential contribution of peripheral inputs from the injured nerve to preservation of the phantom hand map, we tested an upper-limb amputee suffering from brachial plexus avulsion injury with vivid kinaesthetic phantom digit sensations. This injury involves the tearing of the nerve from its attachment at the spinal cord, abolishing afferent inputs and efferent outputs to the residual limb. 3 tesla MRI was used here due to safety restrictions, meaning that standard field digit preference maps were acquired. The bimanual travelling wave task (**A**, *Figure 2*) elicited maps in the controls with reversed colour gradients (**B** - left 2 panels). In comparison to the ultra high-field left hand maps (right panel), the standard field left hand maps were cruder (e.g. less digit specificity, **B** - middle panel). (**C**) The brachial plexus amputee showed two maps with reversed colour gradients, comparable to those found in controls. The map in the missing hand hemisphere showed preference for digits in primary somatosensory cortex. (**D**) Inter-digit representational similarity was assessed for multivoxel patterns underlying the missing hand map. The correlation value of the brachial plexus amputee's inter-digit overlap pattern with the controls exemplified in *Figure 3* fell within the normal controls-to-controls correlation range. Together, these results suggest that preserved representation of the missing hand existed independently of peripheral inputs. Other annotations are as in *Figure 1*.

reorganisation of inputs, rather than reorganisation within S1 itself. This recent evident nicely complements our own finding of SI reorganisation, overlaid on preserved structure and function.

Which inputs could contribute to the maintenance of the missing hand topography? The variability in the level and nature of amputations in this study's cohort allows us to consider the potential contribution of the peripheral nervous system in the preservation of missing hand topography. In the below elbow amputee, some forearm muscles normally controlling hand movements are spared and therefore proprioceptive inputs relating to phantom hand movements likely persist (*Nyström and Hagbarth, 1981*). In the above elbow amputee, these inputs would be absent, though ectopic firing from the injured nerve (*Nyström and Hagbarth, 1981*) or intact dorsal root ganglia could preserve some afferent inputs (as previously shown following peripheral nerve injury [*Nordin et al., 1984*], see also [*Vaso et al., 2014*] for related findings). However, in the amputee suffering from brachial plexus avulsion injury, the dorsal root ganglia are damaged, meaning that no such peripheral input

should be available. Given the observation of preserved topography in all three cases, it is highly probable that the preserved missing hand maps are not maintained by peripheral input, but rather are driven by processing in the central nervous system itself.

What neural signals may be triggering the brain activations subserving the missing hand maps? Given the relatively unimpaired motor system in amputees, it is possible that these representations are driven by motor (efferent) information. The motor system is thought to provide information about its descending commands to the sensory system, by means of efference copy. When efference signals reach the sensory areas, they evoke activity in those areas. The pattern of this corollary discharge could resemble that of the sensory feedback to be expected from the movement (*London and Miller, 2013*). While predictive signals are fundamental components for current theories of motor control (*Franklin and Wolpert, 2011*), surprisingly little empirical evidence exists to demonstrate efferent signals in the primates' S1 hand representation independently of afferent processing (*London and Miller, 2013*). Our evidence for digit topography in S1 despite the physical absence of a hand suggests the involvement of non-afferent processing in S1. The persistence of efference signals from the motor system could contribute to the maintenance of preserved information content in SI despite afferent input loss.

Our findings are based on the unique phenomenology of phantom sensations, and as such all tested amputees reported experiencing exceptionally vivid phantom sensations, allowing them to voluntarily move each of their phantom fingers. An open question remains whether a relationship exists between the experience of phantom sensations and preserved missing hand topography. Our previous findings, showing that phantom hand movements activate the missing hand territory in individuals experiencing varying levels of phantom sensation vividness (*Makin et al., 2013a*; *2015*) might indicate that this is a general phenomenon. However, we note that these previous studies do not provide information on whether the topographic features underlying the phantom-evoked activity were preserved. Further research is needed to determine whether the preservation of missing hand topography depends on (or gives rise to) the experience of phantom sensations, or whether our finding reflects a fundamental organising principle of the brain that is independent of experience.

The notion of brain reorganisation has posed an unexpected obstacle to recent technological developments of human-technology fusion, i.e. neuroprosthetics. Neuroprosthetics allow patients with sensorimotor impairments (e.g. amputation, paralysis) to directly interface with a robotic limb using cortical signals of the hand. However, the relevant information necessary for intuitive neuroprosthesis control may not be attainable once input is lost and functional reorganisation occurs. By demonstrating persistence of topography despite input loss, our finding could be exploited to develop intuitive fine-grained control of neuroprosthetics (*Bensmaia and Miller, 2014*) (e.g. requiring representation of individual digits) in disabled populations.

## Materials and methods

### Participants

The main inclusion criteria for amputees in the study were: (a) unilateral upper limb amputation; (b) vivid kinaesthetic sensations during voluntary movements of each of the five digits of the phantom hand (based on self-report); (c) compatibility with magnetic resonance imaging (MRI) safety guidelines. We initially screened 63 unilateral upper-limb amputees, of which 22 met the initial criteria for MRI safety. Eight amputees met our $2^{nd}$ criterion for vividness of kinaesthetic sensations while volitionally moving each of their phantom digits. Only two of these candidates were approved for undergoing ultra high-field (7 tesla) MRI, based on institute's standard operating procedures for MRI safety. (Ultra high-field (7T) imaging is a newly emerging technique, and as such most clinical implants (e.g. surgical clips) have not been approved for safety in these devices. Therefore, according to current local guidelines it is unsafe to scan individuals who have previously undergone major surgery (e.g. amputees) without a conclusive surgical history.) These amputees were invited to take part in the main study (see *Figure 1—source data 1, 2*, for demographic and clinical details). This involved studying movements of the phantom and intact hands, across two scanning sessions (see *Supplementary file 1*). Two additional inclusion criteria were used for a further investigation of peripheral contributions to the missing hand digit topography: (d) brachial plexus avulsion injury; (e) abolishment of afferent inputs and efferent outputs (based on behavioural and electromyography

(EMG) testing, see below for details). Three amputees who met criteria a, b and c also met criterion d. Only one of these candidates also met criterion e and was therefore invited to participate in the additional study (see *Figure 1—source data 1*, *2*). Because this individual was not approved to undergo ultra high-field MRI, he was tested in a single session at standard field MRI. All three amputees included in the study were amputated on their left arm due to a trauma and were right-hand dominant prior to the amputation (based on self report).

In addition, a group of thirteen two-handed control participants with a dominant right hand were recruited to take part in the study (mean age ± s.e.m. = 43 ± 3; three females). All control participants performed tasks with their left (non-dominant) hand as a control for the phantom digit movements, in a single session. A subset of six age- and sex-matched control participants (mean age ± s.e.m. = 50 ± 14; all male) also completed tasks involving right (dominant) hand movements. One of these participants was subsequently excluded from the study due to an atypical functional digit layout of both hands with a reversed representation of the thumb and index finger. Another participant was excluded due to excessive head motion inside the scanner (greater than the functional voxel dimension (1.2 mm) of absolute mean displacement), leading to visible spin history artefact. The resulting subset of four control participants (45 ± 13; all male: C1 – C4) also completed a standard field (3 tesla) version of the study. Participants in this group therefore completed three sessions (two ultra high-field sessions and one standard field session; see *Supplementary file 1*). Ethical approval was granted by the NHS National Research Ethics service (10/H0707/29) and written informed consent was obtained from all participants prior to the study.

An additional standard field imaging dataset of 15 control participants and the three amputees specified above was also utilised in the current study. The purpose of this analysis was to explore inter-hemispheric (a)symmetry of the intact hand. The pertinent details relating to this dataset are highlighted in *Figure 2—figure supplement 1*, and will not be detailed further in the materials and methods section.

## Behavioural testing

To investigate topographic mapping in the absence of peripheral input, we screened participants with a brachial plexus avulsion injury. This injury involves the tearing of the nerve from its attachment at the spinal cord, abolishing afferent inputs and efferent outputs to the residual limb. To determine the extent of the brachial plexus avulsion injury, we conducted behavioural testing to measure stump sensitivity. Stimuli ranging from light touch to painful pinpricks were applied on the triceps of the residual arm (stump) and a corresponding site on the intact arm. Participants were blindfolded and asked to detect the stimuli, which were presented at varying inter-stimulus intervals in the following order: air puff, light brush, cotton swab stroke, PinPrick probes (MRC systems) with weights: 8 mN, 64 mN and 512 mN. Each stimulus was presented several times. These stimuli are routinely used for qualitative sensoric testing (QST), a technique used to determine sensitivity to touch and pain (*Rolke et al., 2006*).

To assess stump muscle activity we used surface electromyography (EMG). We targeted the biceps and triceps of the residual arm, known to show muscle activity during phantom hand movements in amputees (*Raffin et al., 2012a*).

All three amputees were able to detect each of the stimuli when presented on their intact arm. Only one of the three tested amputees reporting to suffer from brachial plexus avulsion injury was unable to detect any of the stimuli on the residual arm. This participant also did not reveal a change in the EMG channels induced by cued phantom hand movements. This amputee therefore met our requirement for abolishment of afferent inputs and efferent outputs and was included in the standard field (3T) study.

## MRI tasks

Digit representation was probed using a visually cued active (motor) task. In an intact sensorimotor system, movement recruits a combination of peripheral receptors encoding a range of somatosensory modalities (e.g. surface and deeper mechanoreceptors; proprioceptors), as well as efference information from the motor system. Using an active task, we have previously shown high consistency of primary somatosensory (S1) digit topography across multiple scanning sessions (*Kolasinski et al., 2016*).

The present study involved functional MRI (fMRI) paradigms designed to identify the two princi- ples of topographic mapping of the hand (*Graziano and Aflalo, 2007*), as previously shown using neuroimaging: digit selectivity (*Kolasinski et al., 2016*) and inter-digit overlap (*Ejaz et al., 2015*). (1) To identify voxels showing digit preference (digit maps) a travelling wave approach was employed (also known as phase-encoding fMRI) (*Wandell et al., 2007*). This approach has previously been well-validated for sensory body mapping (*Kolasinski et al., 2016*; *Mancini et al., 2012*; *Orlov et al., 2010*; *Sanchez-Panchuelo et al., 2010*; *Sereno and Huang, 2006*; *Zeharia et al., 2015*). Whereas the travelling wave approach provides detailed information regarding digit preference, the winner- takes-all analysis (see below) makes this approach insensitive to overlapping cortical digit represen- tations. Moreover, as the paradigm involves continuous cycles of digit movements with no interrup- tion of baseline periods, regions activated by multiple digits will be difficult to identify using a standard GLM (*Besle et al., 2013*). (2) To identify the extent of overlap between representations of individual digits (inter-digit representational similarity) a block design was therefore also employed. Below we describe the experimental design and analysis in further detail.

Participants were presented with five white circles, corresponding to the five digits, shown on a visual display projected into the scanner bore. To cue the participant which digit should be moved, the circle corresponding to this digit changed (i.e. in colour or by flashing). Participants were instructed to perform individual digit movements with either their intact or phantom hand. Volitional (and not necessarily painful) movement of a phantom hand elicits both central (*Makin et al., 2013a*; *Raffin et al., 2012b*; *Makin et al., 2015*; *Reilly and Sirigu, 2008*) and peripheral motor signals (*Reilly et al., 2006*; *Raffin et al., 2012a*; *Reilly and Sirigu, 2008*) that are different from those found during imagined movements. It was clearly stated to the amputees that they were required to per- form actual movements with their phantom digits (i.e. try and move the digits of the missing hand), rather than motor imagery. To ensure good understanding of these instructions, the amputees were asked to demonstrate to the experimenter outside the scanner the extent of volitional movement carried out in each of their phantom digits, by mirroring each movement onto their intact hand. For the below elbow amputee, stump muscles were palpated by the experimenter outside the scanner to verify that actual movements were executed during movement of the phantom digits.

To compare between the amputees and controls, the phantom (left) hand was matched to the non-dominant (left) hand of control participants, and the intact (right) hand was matched to the (right) dominant hand of controls. Below we describe the parameters used for the main (ultra high- field) study. Adjusted parameters used for the standard field control experiment are mentioned when relevant.

## Missing hand unimanual map

The traveling wave paradigm, as detailed in *Figure 1—figure supplement 1*, involved unimanual digit movements in a set sequence. The task consisted of blocks of 9 s in which the participant was instructed to move one digit. Each digit block was followed by a subsequent block of a neighbouring digit, and repeated as follows: The forward sequence cycled through the movement blocks for dig- its: D1-D2-D3-D4-D5. The backward sequence cycled through the movement blocks in a reverse of the forward sequence (D5-D4-D3-D2-D1, *Figure 1—figure supplement 1A*). A sequence was repeated 8 times per run, with a duration of 6 min and 25 s. A forward and backward sequence was employed in separate runs. Each of the forward and backward runs was repeated twice, with a total duration of 25 min and 40 s.

When one of the white circles on the screen turned red, participants performed self-paced flexion and extension phantom (or left hand) digit movements with the digit corresponding to that circle, for the duration of the block (indicated by the presence of the red circle). The leftmost circle corre- sponded to D5 and the rightmost circle corresponded to D1 of the phantom (or left) hand. Ampu- tees were instructed to move their phantom digits at a comfortable pace. Controls were instructed to perform slow movements, roughly corresponding to the pace and range of movement reported by the amputees (3-4 s per flexion and extension movement). Participants practiced the movements extensively with an experimenter outside the scanner prior to the scan.

### Missing hand inter-digit overlap

To assess inter-digit representational similarity of the missing hand, a block design was also employed. This task involved individual digit movement blocks for each of the five digits, as well as a no movement (rest) condition. Each of these 6 conditions was repeated 7 times in a counterbalanced order and each block lasted 12 s, with a total duration of 8 min and 24 s. Participants were instructed to perform self-paced individual digit movements when the circle corresponding to this digit changed in colour, as described above. All five circles remaining white, and a brief flash of the word 'Rest' indicated the rest condition.

### Bimanual digit map

To replicate the missing hand maps using a separate dataset and additionally test whether topographic representation of the missing hand in amputees was independent of the intact hand, we asked our participants to perform a bimanual travelling wave task. Participants were provided with the visual display of five white circles (as described above) and were instructed to engage both hands in paired simultaneous digit movements when the relevant circle turned red, resulting in reversed digit movement cycles between the two hands. The leftmost circle corresponded to D5 of the left hand and D1 of the right hand; the second left circle corresponded to D4 of the left hand and D2 of the right hand; etc. (*Figure 2A*). A forward and backward sequence was used in separate runs. Each of the forward and backward runs was repeated twice, for a total duration of 25 min and 40 s.

## MRI acquisition

### Ultra high-field

Ultra high-field fMRI data was acquired using a Siemens 7 tesla (7T) Magnetom system with a 32-channel head coil. Task fMRI data was acquired using a limited field of view (FOV), with 19–22 true axial slices centred on the anatomical location of the hand knob (*Yousry et al., 1997*) in the central sulcus bilaterally. The following acquisition parameters were used: sequence: multislice gradient echo EPI, TR: 1500 ms, TE: 25 ms, flip angle: 90°, GRAPPA factor: 2. The spatial resolution was 1.2 mm isotropic.

To improve image registration, a whole brain and a partial field of view single volume high-saturation EPI image were acquired with the same slice positioning as the task fMRI. Anatomical T1-weighed scans, used for surface projection, were acquired using a 3 tesla (3T) system when available. For control participants 1, 5, 6, and 11 a 7T whole-brain T1-weighted image was acquired.

### Standard field

Standard field MRI images were acquired using a 3T Verio MRI scanner (Siemens, Erlangen, Germany) with a 32-channel head coil. A multiband T2*-weighted pulse sequence with an acceleration factor of 6 was used (*Moeller et al., 2010*; *Uğurbil et al., 2013*). This provided the opportunity to acquire data with increased spatial (2 mm isotropic) and temporal (TR: 1300 ms) resolution than available with standard EPI sequences. The following acquisition parameters were used: TE: 40 ms; flip angle: 66°, 72 transversal slices. A high-saturation first volume of each acquired multiband run was collected for registration purposes. Field-maps were acquired for field unwarping.

## MRI analysis

MRI analysis was implemented using tools from FSL and Connectome Workbench software (http://fsl.fmrib.ox.ac.uk/fsl; http://www.humanconnectome.org) (*Jenkinson et al., 2012*; *Woolrich et al., 2009*; *Smith et al., 2004*) in combination with in house scripts developed using Matlab (version 8.4, R2014b). Cortical surface reconstructions, used for visualisation of the fMRI results, were produced using FreeSurfer (http://freesurfer.net) (*Dale et al., 1999*; *Fischl et al., 2001*).

### MRI preprocessing

Common pre-processing steps for fMRI data were applied to each individual run in native (three dimensional, 3D) space, using FSL's Expert Analysis Tool FEAT (v6.00; fsl.fmrib.ox.ac.uk/fsl/fslwiki). The following steps were included: Motion correction using MCFLIRT (*Jenkinson et al., 2002*), brain extraction using automated brain extraction tool BET (*Smith, 2002*), spatial smoothing using a

1.5 mm FWHM (full width at half maximum) Gaussian kernel for the ultra high-field scans and 2 mm FWHM for the standard field scans, and high pass temporal filtering with a cut-off of 100 s. All BOLD EPI data were assessed for excessive motion using motion estimate outputs from MCFLIRT: ultra high-field functional data from one participant exhibited greater than 1.2 mm (functional voxel size) of absolute mean displacement and was excluded from all further analysis.

## Image registration
### Ultra high-field
Image registration was accomplished using FLIRT (FMRIB's linear image registration tool) (*Jenkinson et al., 2002*; *Jenkinson and Smith, 2001*). To ensure good registration of the partial-FOV functional data to the anatomical image, image registration was carried out in individual, visually inspected, steps.

First, the task fMRI data from each run was registered to a partial-FOV single volume high-saturation EPI image, acquired at the first session. This partial-FOV image was registered to a whole brain FOV single volume high-saturation EPI image. The whole-brain image was then registered to the T1-weighted image; initially using the mutual information cost function (6 degrees of freedom), and then optimised using boundary-based registration (*Greve and Fischl, 2009*) (6 degrees of freedom, FMRIB's Automated Segmentation Tool (FAST) for white matter segmentation, no search). For two participants (C4 and C11), manual alignment was used to register the single volume partial-FOV high-saturation EPI image to the structural white matter and pial surfaces using blink comparison as implemented in Freeview.

### Standard field
Functional and anatomical images were aligned using a similar pipeline as described above. Each individual run was first co-registered to the high-saturation first volume of the acquired multiband run, and then to the T1-weighted image. In addition, individual field-maps and field-map based unwarping of the multiband images were included to reduce spatial distortions and additionally improve co-registration.

### Travelling wave analysis
The travelling wave runs were analysed for each individual participant in native (3D) space, using a cross-correlation approach previously applied in retinotopy (*Engel et al., 1997*). The approach is based on continuous presentation of stimuli in a set cycle that are expected to result in neighbouring cortical representations (e.g. rotating wedge in the visual field; sequential digit stimulation in a set cycle). It is designed to capture voxels showing increased response to one condition, above and beyond all other conditions (in our case, preference for a specific digit). Beyond preference, this technique also provides a powerful tool for capturing the smooth progression of adjacent representations that are typical for topographic maps. It is therefore considered a preferable technique for capturing topographic representations (*Wandell et al., 2007*). With respect to somatotopy, the travelling wave approach (or the homologous phase-encoding approach) has previously been validated against blocked and event-related fMRI paradigms (*Kolasinski et al., 2016*; *Orlov et al., 2010*; *Sanchez-Panchuelo et al., 2010*; *Besle et al., 2013*).

We have recently utilized the traveling wave approach, in combination with ultra high-field fMRI to demonstrate highly reproducible maps of individual digits in S1 (*Kolasinski et al., 2016*). In the current study, we closely followed these previously validated experimental procedures, as described in *Figure 1—figure supplement 1*. Participants moved individual digits in a set cycle (see the fMRI tasks section above). A reference model was generated using a gamma-HRF convolved boxcar function, while taking into account the hemodynamic delay. The model was constructed using a 9 s 'on' (the duration of a single digit movement) and 36 s 'off' (the period of movement of all other digits), accounting for a single 45 s cycle (*Figure 1—figure supplement 1B*). This cycle was repeated 8 times to reflect the full run duration. The reference model was systematically shifted in time to model activity throughout the full movement cycle. Because the runs were acquired using a TR of 1.5 s, the model was shifted 30 times by 1 lag to account for the full 45 s cycle of movement.

For each individual voxel, each of the 30 reference models was correlated with the preprocessed BOLD signal time course to estimate cross-correlation values. The resulting r-values were

standardised using the Fisher's r-to-z transformation. By plotting these standardised r-values as a function of the lag, tuning curves can be created, and the optimal fit for each voxel can be inferred (*Figure 1—figure supplement 1C*).

The travelling wave approach uses a set cycle, and as such could be susceptible to order-related biases resulting from the sluggish hemodynamic response. For this reason, the order of the cycle was varied between forward (D1-D2-D3-D4-D5) and backward (D5-D4-D3-D2-D1) in different runs (*Figure 1—figure supplement 1A*). To average across the four runs, lags were initially assigned to each of the five digits (six lags per digit). Within each voxel, the r-values corresponding to each digit were averaged, resulting in five r-values, corresponding to each of the digits for a given run. The digit-specific r-values were then averaged across the forward and backward runs on a voxel-by-voxel basis. A winner-take-all approach was applied to produce maps in which each voxel was assigned exclusively to one individual digit, providing us with digit specificity.

To visualise the gradient of progression across digits, lag-specific maps were also produced. For each backward run, the resulting r-values from the cross-correlation analysis were standardised and time-reversed. Forward and backward runs were averaged, on a lag-by-lag basis. To construct a gradient map, a winner-take-all approach was used across all 30 lags, in which each voxel was assigned exclusively to one individual lag.

Cortical surface projections were constructed from T1-weighted images. The digit and gradient maps were registered to structural space and projected to two-dimensional surface space using a cortical ribbon mapping method. To account for multiple comparisons, thresholding was implemented on the surface using the false discovery rate (FDR) (*Benjamini, 1995*), calculated for each digit individually. This approach doesn't take into consideration the neighbourhood relationships of the voxels, and therefore doesn't force cluster patterns. The thresholded maps were set at a false detection criterion of q<0.05 based on the native (3D) values. The FDR thresholded digit-specific clusters were overlaid into a single hand map. Within the resulting five-digit map, we used a 30-lag colour code to visualise the gradient of progression across digits (*Figure 1—figure supplement 1D*).

For the standard field digit maps, the same procedures were used, with the following exceptions. Due to the difference in temporal resolution (TR = 1.3 s), each digit block lasted 9.1 s, resulting in slightly longer movement cycles (45.5 s). 35 HRF reference models were constructed (compatible with 35 lags), and a more conservative FDR criterion (q<0.01) was selected, due to a reduced signal/noise ratio.

## Split-half consistency

To quantify consistency in digit preference, as identified using the travelling wave task, we split the data used to construct the digit preference maps, and compared the spatial correspondence between digit selective clusters. Split-half consistency of the digit-specific clusters was calculated using the Dice coefficient (*Kolasinski et al., 2016*; *Dice, 1945*). The Dice coefficient varies from 0 (no spatial correspondence between digit representations) to 1 (perfect spatial correspondence between digit representations). Where A and B are the areas of two digit representations, the Dice Coefficient is expressed as:

$$\frac{2 \times |A \cap B|}{|A| + |B|}$$

Each digit's winner-take-all map was minimally thresholded on the cortical surface (Z>2). Spatial correspondence was calculated in S1 (as defined by FreeSurfer) between each possible digit pair across the split-halves of the unimanual digit maps (*Figure 1*, *Figure 1—figure supplement 3*). The first forward and backward runs were combined to provide the first digit-specific clusters, and the second forward and backward runs were combined to form the second digit-specific clusters. By doing so, we established a split-half consistency measure for the travelling wave maps of the phantom (or left) hand and intact (or right) hand separately. One control participant was discarded from the subsequent analysis, as his mean spatial correspondence values (across all possible digit combinations) fell outside two standard deviations from the control group mean.

A benchmark for digit selectivity is greater split-half spatial correspondence across 'same' versus 'different' digits. Mean 'same'-digit values (across all five 'same'-digit pairs) were greater than mean

'different'-digit values (across all twenty 'different'-digit pairs) in each of the participants. The 'same'-digit spatial correspondence values in the two amputees (0.30 for the above and below elbow amputees, average across all 'same'-digit pairings) were smaller than those observed in the control participants (0.57 on average across all control participants; *Figure 1D*), suggesting lower consistency of digit selectivity in amputees. To further assess digit selectivity in the amputees, the 'different'-digit distribution of Dice spatial correspondence values was estimated using a bootstrapping procedure (5000 iterations) for each individual participant. 'Same'-digit values were averaged for each participant, and compared against the 95% confidence interval (CI) on a subject-by-subject basis. For all participants (amputees and controls), the 'same'-digit value fell well outside the range of spatial correspondence seen between 'different' digits. This indicates that the consistency shown for digit-selective clusters was greater than expected by chance in all participants. Next, the mean 'same'-digit values in the two amputees were compared to the 'different'-digit spatial correspondence distribution of each of the controls. 'Same'-digit spatial correspondence values in both amputees fell outside the 'different'-digit spatial correspondence distribution of each of the controls (CI: 0.03–0.08 on average). This was also confirmed by comparing the mean spatial correspondence between 'same'-digit values for each of the amputees and 'different'-digit values of the controls (two-tailed Crawford and Howell t-test [*Corballis, 2009*]; $t_{(9)}$ = 14.24, p<0.001 and $t_{(9)}$ = 14.31, p<0.001 for above and below elbow amputees respectively). This indicated that although reduced, the digit maps of the amputees were consistent, even relative to controls.

In addition we aimed to compare between the unimanual and bimanual digit maps within the missing hand hemisphere (*Figure 2E*), providing us with an estimate of consistency when accounting for potential contributions of the intact hand. The 'different'-digit distribution of Dice spatial correspondence values between the unimanual and bimanual maps was estimated using a bootstrapping procedure (5000 iterations) for each individual participant. 'Same'-digit spatial correspondence values (mean across all 'same' digits) in both amputees (0.49 and 0.40 for above and below elbow amputees respectively) fell outside the 'different'-digit spatial correspondence distribution. This was observed both compared to their own 'different'-digit distribution (CI: 0.03–10 for both above and below elbow amputees) and each of the controls' 'different'-digit distributions (CI: 0.03–0.13 on average). This was also confirmed by comparing the mean spatial correspondence between 'same'-digit values for each of the amputees and 'different'-digit values of the controls (two-tailed Crawford and Howell t-test [*Corballis, 2009*]; $t_{(3)}$ = 29.14, p<0.001 and $t_{(3)}$ = 22.57, p<0.001 for above and below elbow amputees respectively).

## Inter-digit representational similarity analysis

We used an fMRI pattern component approach to identify the extent to which representations of the different digits overlapped with each other, while considering multivoxel patterns underlying the S1 digit maps. Voxels underlying the digit preference map surrounding the anatomical hand knob were combined to form a region of interest (ROI) in native space. The multivoxel pattern-component model is described in detail in (*Diedrichsen et al., 2011*) and implemented in http://www.icn.ucl.ac.uk/motorcontrol/imaging/multivariate_analysis.html. This method allows deriving unbiased estimates of the true correlations between the underlying activation patterns for each condition.

In brief, a generative model is created that assumes the observed patterns are associated with a set of underlying pattern components that relate to the different experimental conditions or noise. Activity patterns are decomposed within the ROI into a common (noise) component that is shared between all trials and one specific component for each of the five digits. By estimating the variability (or strength) of the similarity between the latter components, the inter-digit representational overlap in the ROI is revealed. *Figure 3* and *Figure 3—figure supplement 1* (*Figure 4D* for standard field) show the resulting matrices, demonstrating the level of representational similarity (or overlap) between each digit pair. The topographic inter-digit pattern is characterised by increased overlap between neighbouring digits. Furthermore, digits showing increased synergies in daily life (e.g. digits 3 and 4) tend to show greater representational overlap than digits that are more independent during hand function (e.g. digits 1 and 2) (*Ejaz et al., 2015*). The resulting inter-digit representational overlap pattern therefore provides a detailed 'fingerprint' of hand representation.

We initially assessed overall (mean) overlap within the matrix of each participant, by averaging all

unique inter-digit similarity r-values (across the 10 cells of the matrix, while excluding the diagonal). On average, the amputees showed higher inter-digit overlap (0.55 and 0.54 for above and below elbow amputees, respectively) than the control group (CI: 0.24–0.41). To investigate whether the missing hand inter-digit overlap pattern was comparable to normal hand representation, as found in controls, we generated the distribution (and 95% confidence intervals) of correlation across the control population. For this purpose, inter-digit overlap patterns (i.e. the cells in the similarity matrices) were correlated across pairs of controls, using a bootstrapping approach (10,000 iterations) and a Spearman test. One control participant was discarded from the subsequent analysis, as his mean Spearman correlation value (with all other controls) fell outside two standard deviations from the control group mean. (Note that the exclusion of the outlier participant did not affect the outcome of this analysis.) Next, the same procedure was carried out between each amputee and each of the controls' inter-digit overlap patterns. By plotting the average amputee-control Spearman rho values and confidence intervals against the confidence interval of the controls-to-controls correlation distribution, we were able to show that the amputees' inter-digit representational overlap pattern fell within the normal range of the controls. There was no significant difference between the amputees-to-controls average Spearman correlation and the control-to-control correlations also when using more conventional statistics (two-tailed Crawford and Howell t-test [*Corballis, 2009*]; $t_{(9)} = -1.10$, p=0.30 and $t_{(9)} = 0.31$, p=0.76 for above and below elbow amputees respectively).

Because the pattern component approach implemented here is minimally affected by differences in noise amplitude resulting from different scanners and acquisition parameters (*Diedrichsen et al., 2011*), we repeated the same analysis, using the brachial plexus amputees' data and the control participants' data (ultra high-field MRI). The average correlation of this amputee's inter-digit overlap pattern with each of the controls' inter-digit overlap patterns (0.76), fell within the controls-to-controls correlation range. There was no significant difference between the amputee-to-controls average Spearman correlation and the control-to-control correlations (two-tailed Crawford and Howell t-test [*Corballis, 2009*]; $t_{(9)} = 0.14$, p=0.89)

We also calculated a Bayesian t-test to compare between the controls-to-controls distribution and each of the amputee-to-controls distributions using JASP (*Morey*; *Love*; *Rouder et al., 2009*). The Cauchy prior width was set at 0.707 (default). Based on the well accepted criterion of Bayes factor smaller than 1/3 (*Wetzels et al., 2011*; *Dienes et al., 2014*) our findings support the null hypothesis (amputees' inter-digit similarity pattern is not different from controls) for the below elbow and brachial plexus amputee (Bayes factor = 0.23 and 0.28 respectively). For the above elbow amputee the evidence was inconclusive (Bayes factor = 2.94). Note that this criterion for Bayesian factor is considered as moderate (though positive) evidence (*Kass et al., 1995*).

## Acknowledgements

The study was funded by the Wellcome Trust and the Royal Society. SK is supported by the UK Medical Research Council and Merton College, Oxford. JK holds a Stevenson Junior Research Fellowship at University College, Oxford. SJ is supported by the UK Medical Research Council (MR/L009013/1). CFB is supported by the Netherlands Organisation for Scientific Research (NWO-Vidi 864-12-003) and gratefully acknowledges funding from the Wellcome Trust UK Strategic Award (098369/Z/12/Z). IT is supported by the following: Wellcome Trust Strategic Award and NIHR Oxford Biomedical Research centre. HJB is a Wellcome Trust Principal Research Fellow (110027/Z/15/Z). TRM holds a Sir Henry Dale Fellowship jointly funded by the Wellcome Trust and the Royal Society (104128/Z/14/Z). We thank our participants for taking part in the study. We thank Devin Terhune and Naveed Ejaz for advice on analysis and Tim Vogels, Paul Matthews, Jody Culham, Tim Behrens and Holly Bridge for comments on the manuscript.

## Additional information

### Competing interests
HJ-B: Reviewing editor, *eLife*. The other authors declare that no competing interests exist.

## Funding

| Funder | Grant reference number | Author |
| --- | --- | --- |
| Merton College, University of Oxford | Graduate School Studentship | Sanne Kikkert |
| Medical Research Council | Graduate School Studentship | Sanne Kikkert |
| University College, Oxford | Stevenson Junior Research Fellowship | James Kolasinski |
| Medical Research Council | MR/L009013/1 | Saad Jbabdi |
| Wellcome Trust | Strategic Award | Irene Tracey |
| NIHR Oxford Biomedical Research Centre | | Irene Tracey |
| Nederlandse Organisatie voor Wetenschappelijk Onderzoek | NWO-Vidi 864-12-003 | Christian F Beckmann |
| Wellcome Trust | UK Strategic Award, 098369/Z/12/Z | Christian F Beckmann |
| Wellcome Trust | Principal Research Fellow, 110027/Z/15/Z | Heidi Johansen-Berg |
| Royal Society | Sir Henry Dale Fellowship, 104128/Z/14/Z | Tamar R Makin |
| Wellcome Trust | Sir Henry Dale Fellowship, 104128/Z/14/Z | Tamar R Makin |

The funders had no role in study design, data collection and interpretation, or the decision to submit the work for publication.

## Author contributions

SK, JK, TRM, Conception and design, Acquisition of data, Analysis and interpretation of data, Drafting or revising the article; SJ, IT, CFB, Analysis and interpretation of data, Drafting or revising the article; HJ-B, Conception and design, Analysis and interpretation of data, Drafting or revising the article

## Author ORCIDs

James Kolasinski, http://orcid.org/0000-0002-1599-6440
Tamar R Makin, http://orcid.org/0000-0002-5816-8979

## Ethics

Human subjects: Ethical approval was granted by the NHS National Research Ethics service (10/H0707/29) and written informed consent was obtained from all participants prior to the study.

# Additional files

## Supplementary files

• Supplementary file 1. Runs acquired for each participant.

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
