## [Decision Letter]

Thank you for submitting your article "Revealing the neural fingerprints of a missing hand" for consideration by *eLife*. Your article has been reviewed by three peer reviewers, one of whom is a member of our Board of Reviewing Editors, and the evaluation has been overseen by Sabine Kastner as the Senior Editor. The following individuals involved in the review of your submission have agreed to reveal their identity: Klaas Enno Stephan (Reviewing Editor and Reviewer #1) and Brigitte Röder (Reviewer #2).

The reviewers have discussed the reviews with one another and the Reviewing Editor has drafted this decision to help you prepare a revised submission.

Summary of manuscript:

This paper revisits the question to what degree sensory representations in cortex are plastic following the loss of certain sensory inputs. Two amputees who lost their hands were studied with 7 Tesla fMRI, using a paradigm that cued phantom digit movements. An additional amputee with brachial plexus avulsion was investigated using 3 Tesla fMRI. Healthy controls served as comparison. The authors report that a representation of the missing hand's fingers can be detected in the primary somatosensory cortex of amputees, even though their loss of limb dates back several decades.

Summary of reviews:

All three reviewers were positively impressed by the paper and thought that it contributes some very interesting findings would usefully inform our understanding of plasticity of sensory representations in cortex. They agreed that the experimental design and the statistical analyses were of excellent quality. However, they also had a number of comments and suggestions which would need to be addressed in a revision of your paper.

The policy of the journal is to provide you with a single set of comments which reflect the consensus view amongst reviewers. These comments can be found below and are divided into essential or Major Issues, which must be addressed convincingly, and Minor Issues. We hope that you will find these comments helpful to further improve the paper.

Essential revisions:

1) All reviewers felt that the conclusions were too strong – both considering the strength of the statistical results and with regard to their neuroscientific framing – and should be toned down and nuanced. In brief, the reviewers think it would be appropriate to conclude that hand representations are still detectable decades after amputation but the impression should be avoided that amputees cannot be distinguished from controls. This is detailed by the next two points.

2) With regard to statistics, the results of the analyses are more mixed and less decisive than the conclusions suggest. Concerning digit selectivity, there is a qualitative correspondence of selectivity maps between amputees and controls, but the selectivity is significantly reduced in amputees. Concerning the second part of the analysis, interdigit overlap, some of the tests also indicate significant differences between amputees and controls. Furthermore, the results of the Bayesian analysis are relatively weak; this should be acknowledged more clearly. Two of the Bayes factors reported are in favour of the authors' hypothesis of no difference between groups, but are just very slightly above the conventional threshold for "positive evidence" (BF <0.33 or >3, according to the Kass & Raftery classification); a third Bayes factor reported goes against their hypothesis, but marginally fails to reach the positive evidence threshold. Notably, in Bayesian statistics, the positive evidence threshold is not usually considered sufficient to disambiguate hypotheses decisively; for this, one would typically require a Bayes factor of at least 20.

3) Concerning the neuroscientific aspects, the reviewers thought that the historical framing and interpretation of the findings should be approved. They point out that, had these findings been published 30 years ago, they would not have been regarded as surprising. It was not before the work of M. Merzenich in 1980 that evidence became available against the "hard-wired dogma" of adult brains. However, this has not been fully replaced with a "soft-wired dogma" since then; rather, it is widely recognised that both aspects coexist. While it has become clear that the brain has considerable capacity for functional reorganisation and plasticity even in adult life (for example, concerning somatotopy, electrophysiological work has demonstrated that neurons are partially able to acquire new receptive fields after deafferenting a digit etc.), there are hardly claims or data suggesting a full loss of digit somatotopy. By contrast, some authors have claimed that individual digit topography can be detected by touching the face of some amputees. In brief, reorganization and preserved digit somatotopy are not mutually exclusive, but reflect a balance of stability and plasticity in the CNS. The reviewers think that this is exactly what the statistical results reflect (see point 2 above) and find that a more balanced interpretation and discussion is necessary. The present conclusions (including the abstract) oversimplify and are worded too strongly.

4) All patients selected reported "vivid kinaesthetic sensations". This raises the question whether those amputees reporting vivid sensations of their digits are those who maintain somatotopy, while the others do not. In other words, the present results may be specific for a particular subgroup of amputees; this should be discussed.

5) How well did the patients manage to move individual digits, and how can this be assessed, given that this movement was imaginary?

6) The complexity of the possible neurophysiological processes that underlie the fMRI results requires more consideration in an extended discussion, specifically with regard to the possible pathways that might have generated the brain activity observed here. The following three comments may be useful in this regard:

a) Equally as interesting as the preservation of the deafferented glabrous finger map in area 3b and 1, per se, is fact that these maps could be selectively locally activated, presumably by higher level postcentral somatosensory areas and/or by precentral motor areas. Understanding the pathway of activation is further challenged by the differences in the 3 cases presented here. In the below-elbow case, it is likely that portions of the extensor digitorum communis (finger raising) and flexor digitorum profundus (finger curling) muscles as well as their afferent and efferent innervation remain and were activated during the 'imagined' movements, which could probably be better described as 'executed' (muscle) movements. In this case, one might imagine the involvement of M-I to S-I connections, along with all the complexities of the difference between muscles located in the arm versus glabrous skin on the underside of the fingers. But in the above-elbow case, those finger muscles in the arm are gone; although dorsal root ganglion cells formerly innervating glabrous finger skin as well as motorneurons formerly innervating those muscles are likely to have persisted above the injury, and may have made ectopic connections. Finally, the brachial plexus avulsion case would likely have involved damage to the dorsal root ganglia and the spinal cord itself. In that case, the most peripheral source of abnormal neural activity could only have been neurons in the dorsal horn itself.

b) An important piece of information to keep in mind when interpreting these results is that there are several levels of map before we reach the cortex: dorsal root ganglion -> dorsal column nuclei -> ventrobasal nucleus -> S-I, and dorsal root ganglion -> dorsal column nuclei -> ventrolateral nucleus -> M-I cortex. Recent experiments on non-human primates with chronic lesions of the dorsal column subserving the arm have suggested that our initial expectations for sites of plastiticy may not have been correct (for example, P Chand and N Jain (2015) Intracortical and thalamocortical connections of the hand and face representations in somatosensory area 3b of macaque monkeys and effects of chronic spinal cord injuries. Journal of Neuroscience 35:13475-13486.) That study showed that despite the electrophysiological presence of large-scale somatotopic reorganization in 3b, surprisingly, there was no evidence for any marked increase in cortico-cortical connections across the hand-face border in 3b, *or* any evidence of altered map connections from the hand and face parts of the ventrobasal nucleus to the cortex. So the large-scale expansion of the chin representation into the deafferented hand representation in 3b may actually instead have been driven by the reorganization of the dorsal column nuclei. In the context of the present paper, this suggests that reorganized inputs to cortical maps should strongly be distinguished from the intrinsic organization of the cortical maps themselves, which contrary to expectations, may turn out to be *less* plastic than more peripheral stations."

c) Another point perhaps worth a sentence in the Discussion is that the experiment performed with the below-the-elbow amputee – having them contract their finger muscles without generating an stimulation of the (missing) glabrous skin surface of the finger is not straightforwardly possible with an intact limb since any activation of muscles that move the fingers will generate some direct somatosensory stimulation, even with the hand held up in the air. However, it might be possible to test this in an intact limb by anesthetizing an intact hand with a wrist block to see if cortical region previously thought to be mainly driven by stimulation of the glabrous surface of the fingers can be directly driven by efference copy (or other) signals.

d) The fact that phantom limb sensations can arise almost immediately during brachial blocks (Gentili ME, Verton C, Kinirons B, Bonnet F. (2002) Clinical perception of phantom limb sensation in patients with brachial plexus block. Eur J Anaesthesiol 19:105-108 also suggests that some 'plasticity' is immediate.

---

## [Author Response]

Essential revisions:

*1) All reviewers felt that the conclusions were too strong – both considering the strength of the statistical results and with regard to their neuroscientific framing – and should be toned down and nuanced. In brief, the reviewers think it would be appropriate to conclude that hand representations are still detectable decades after amputation but the impression should be avoided that amputees cannot be distinguished from controls. This is detailed by the next two points.*

We take the reviewers point that the persistent representation of the missing hand was not entirely matched to the controls’ topography in all tests. For example, results of our analysis indeed suggest that digit selectivity in amputees was weaker than in controls. In the revised manuscript we have reworded the description of our results throughout the text to better reflect our main findings as well as their limitations (please see below for key examples from the Abstract, Impact statement and Discussion). Furthermore, in the revised Discussion we also highlight the extent to which digit representation differs in amputees and controls.

Revised text in the Abstract:

“We show that representation of the missing hand’s individual fingers persists in the primary somatosensory cortex even decades after arm amputation. By demonstrating stable topography despite amputation, our finding questions the extent to which continued sensory input is necessary to maintain organisation in sensory cortex, thereby reopening the question what happens to a cortical territory once its main input is lost.”

Revised text in the Impact statement:

“We show that the functional finger layout of the primary somatosensory cortex, previously thought to depend on experience, remains distinct despite the physical absence of fingers for several decades.”

Revised text in the Discussion of the Main text:

“Together we show that, although the missing hand maps were weaker and noisier than the maps found in controls, the functional digit layout of S1 prevails following arm amputation. Digit topography, previously thought to depend on experience (Ejez, Hamada and Diedrichsen, 2015; Merzenich et al., 1984), was detectable despite decades without organised peripheral inputs associated with normal hand function.”

*2) With regard to statistics, the results of the analyses are more mixed and less decisive than the conclusions suggest. Concerning digit selectivity, there is a qualitative correspondence of selectivity maps between amputees and controls, but the selectivity is significantly reduced in amputees. Concerning the second part of the analysis, interdigit overlap, some of the tests also indicate significant differences between amputees and controls. Furthermore, the results of the Bayesian analysis are relatively weak; this should be acknowledged more clearly. Two of the Bayes factors reported are in favour of the authors' hypothesis of no difference between groups, but are just very slightly above the conventional threshold for "positive evidence" (BF <0.33 or >3, according to the Kass & Raftery classification); a third Bayes factor reported goes against their hypothesis, but marginally fails to reach the positive evidence threshold. Notably, in Bayesian statistics, the positive evidence threshold is not usually considered sufficient to disambiguate hypotheses decisively; for this, one would typically require a Bayes factor of at least 20.*

As indicated in our response to point 1 above, we accept the reviewers’ reservation. We revised the manuscript to better reflect the outcomes of our various analyses, including those showing differences between the amputees and controls. Rather than stating that the amputees show identical topography as controls, we now state that topography remains detectable in amputees.

For revised text in the Abstract, Impact statement and Discussion of the Main text, see response to Reviewers’ point 1.

The interpretation of the results of our Bayesian analysis has been based on the paper by Zoltan Dienes “Using Bayes to get the most out of non-significant results” (Dienes, 2014). According to this paper, a Bayes Factor (BF) under 0.33 provides coherent evidence to support the null hypothesis. This criterion is consistent with the results found in two of the amputees. A BF between 0.33 and 3 should be interpreted as ambiguous evidence (neither confirming or rejecting the null hypothesis), as found in the third amputee. This BF criterion is currently used as a standard in the field. For example, in a key paper by Wetzels et al. (Wetzels et al., 2011), a BF greater than 3 (or lower than 0.33) is considered substantial evidence for (or against) the hypothesis. In this paper, a BF of 3 is shown to correspond to a p value of 0.01. In the guidelines by Kass and Raftery (Kass and Raftery, 1995), where a criterion of BF > 20 is suggested as strong evidence for the alternative hypothesis, a BF greater than 3 is still considered positive evidence (thus validating out interpretation for BF < 0.33). Nevertheless, to correspond to the most stringent criteria, we interpret our findings as “moderate evidence” in favour of the null hypothesis in the revised manuscript.

Revised text in the Materials and methods:

“We also calculated a Bayesian t-test to compare between the controls-to-controls distribution and each of the amputee-to-controls distributions using JASP (Love et al; Morey and Rouder; Rouder et al. 2009). The Cauchy prior width was set at 0.707 (default). Based on the well accepted criterion of Bayes factor smaller than 1/3 (56,57) our findings support the null hypothesis (amputees’ inter-digit similarity pattern is not different from controls) for the below elbow and brachial plexus amputee (Bayes factor = 0.23 and 0.28 respectively). For the above elbow amputee the evidence was inconclusive (Bayes factor = 2.94). Note that this criterion for Bayesian factor is considered as moderate (though positive) evidence (Kass and Raffery, 1995).”

*3) Concerning the neuroscientific aspects, the reviewers thought that the historical framing and interpretation of the findings should be approved. They point out that, had these findings been published 30 years ago, they would not have been regarded as surprising. It was not before the work of M. Merzenich in 1980 that evidence became available against the "hard-wired dogma" of adult brains. However, this has not been fully replaced with a "soft-wired dogma" since then; rather, it is widely recognised that both aspects coexist. While it has become clear that the brain has considerable capacity for functional reorganisation and plasticity even in adult life (for example, concerning somatotopy, electrophysiological work has demonstrated that neurons are partially able to acquire new receptive fields after deafferenting a digit etc.), there are hardly claims or data suggesting a full loss of digit somatotopy. By contrast, some authors have claimed that individual digit topography can be detected by touching the face of some amputees. In brief, reorganization and preserved digit somatotopy are not mutually exclusive, but reflect a balance of stability and plasticity in the CNS. The reviewers think that this is exactly what the statistical results reflect (see point 2 above) and find that a more balanced interpretation and discussion is necessary. The present conclusions (including the abstract) oversimplify and are worded too strongly.*

We fully agree that our results demonstrate that a delicate balance exists between soft- and hard-wiring. We also agree that this view has been prevalent in the field. For example, in the original digit reorganisation paper by Merzenich et al. (Merzenich et al., 1984), the authors suggested that phantom sensations most likely arise from a preserved functional representation of the missing digit – therefore not excluding the potential for persistent organisation co-occurring with reorganisation. However, evidence to support this co-existence of preserved organisation and reorganisation are only now beginning to emerge (see point 7 for further discussion of how the visual community resolves between the hard- and soft-wired dogmas; see also point 6b for further discussion of how reorganisation and preserved representation can co-occur following sensory input loss). We believe that the discrepancy between the theoretical framework, embracing the co-occurrence of preservation and plasticity, and the mainstream evidence in favour of plasticity, is due to methodological limitations. Once the main input is lost to a brain area, scientists resort to studying representations of its cortical neighbours to best characterise the freed-up territory (e.g. adjacent digits, arm or face in primates following digit or arm deafferentation (Merzenich et al., 1984; Pons et al., 1991; Kambi et al., 2014)(1); neighbouring whiskers in rodents following whisker removal (2,3); or peripheral visual representation in humans with foveal visual loss (Baseler et al., 2011)). This approach leaves unexplored the possibility that the original function of the region may be preserved, though latent. Our experimental approach therefore provides a new perspective on the capacity for the brain to maintain hard-wired representation despite a drastic change in inputs. We revised the discussion of the manuscript to better reflect the existing theoretical framework. For example, we discuss previous literature in the visual field suggesting limits of plasticity in the adult human brain (see also point 7) and explain how our research advances on these findings.

Revised text in the Abstract:

“By demonstrating stable topography despite amputation, our finding questions the extent to which continued sensory input is necessary to maintain organisation in sensory cortex, thereby reopening the question what happens to a cortical territory once its main input is lost.”

Revised text in the Discussion of the Main text:

“Conversely, recent research in the visual cortex suggests that reorganisation in the adult brain may be restricted. For example following macular degeneration, the functional representation of the intact visual field was unchanged ((Baseler et al., 2011), see (Smirnakis et al., 2005) for similar results in non-human primates). Common to all these previous studies aiming to characterise reorganisation or the lack thereof, is that the authors probed the cortical neighbours of the area previously responsible for processing the lost input. While this approach is suitable for documenting shifted representation of the cortical neighbours, it leaves unexplored the possibility that the original function of the region deprived of sensory input may be preserved, though latent. Amputees experiencing phantom sensations provide a unique model to study what happens to the deprived cortical territory itself during sensory input loss.”

Further, we fully agree with the reviewers that reorganisation and preserved digit topography are not mutually exclusive. In fact, we believe that the main strength of our results is revealing the latent hard-wired finger maps, despite reorganisation. Two of the three amputees tested in the current study revealed overrepresentation of the intact hand in the missing hand territory, compatible with reorganisation (Figure 2—figure supplement 1; see also point 12). In the revised manuscript we further emphasise and interpret the co-occurrence of reorganisation and preserved missing hand topography in the main text and the Discussion.

Revised text in the Main text:

“This result also demonstrates the independence of the missing hand topography from the representation of the intact hand. Our group and others previously showed that following unilateral arm amputation, the intact hand becomes over-represented in the missing hand territory (Makin et al., 2013; Philip and Frey, 2014; Raffin et al., 2016). Our results therefore confirm that the digit map in the missing hand territory is not driven by an emerging representation of the intact hand. Note, however, that our results do not exclude the possibility for reorganisation in the missing hand territory. As demonstrated in Figure 2—figure supplement 1, this study’s participants showed over-representation of their intact hand in the missing hand territory, indicating that the preserved missing hand topography can co-occur with remapping of body parts.”

Revised text in the Discussion of the Main text:

“How can our finding of preserved S1 topography of a missing hand be allied with the wealth of evidence showing cortical reorganisation in S1 following sensory input loss? (e.g. amputation and spinal cord injury (Merzenich et al., 1984; Pons et al., 1991; Jain et al. 2008); see Figure 1—figure supplement 2 for reorganisation in the current study’s volunteers cohort). Already in their seminal work Merzenich et al. suggested that reorganisation following sensory input loss does not exclude the possibility for simultaneous preservation of the original function of that region (Merzenich et al., 1984). Accordingly, recent structural and functional evidence shows that the capacity for S1 reorganisation is more limited than initially thought, and that instead the functional changes previously observed in S1 following input loss could be attributed to reorganisation in sub-cortical areas in the afferent pathway, principally the brainstem (Jain, Catania and Kaas, 1998; Kambi et al., 2014; Chand and Jain, 2015). In other words, previous findings of massive cortical reorganisation in S1 reflect reorganisation of inputs, rather than reorganisation within S1 itself. This recent evident nicely complements our own finding of SI reorganisation, overlaid on preserved structure and function.”

We decided not to discuss the case studies reporting facial referred sensations (4,5) as potential evidence in favour of the co-occurrence of reorganisation and preserved missing hand representation. This is due to limitations in the original experimental designs (e.g. the studies were not systematically conducted, did not involve control conditions and were not double blind), as well as the qualitative report of the findings (e.g. no statistical quantification was carried out). Indeed, further studies that avoided some of these methodological pitfalls showed that referred sensations could be triggered by touch applied on body parts that are not cortical neighbours of the hand area (e.g. feet, chest and neck), as well as body parts that are contralateral to the missing hand (6, 7). As such, we felt there is insufficient evidence to link referred sensations and SI organisation and/or reorganisation. Nevertheless, we would be happy to discuss these studies should the editor or reviewers feel that this information should be added to the manuscript.

*4) All patients selected reported "vivid kinaesthetic sensations". This raises the question whether those amputees reporting vivid sensations of their digits are those who maintain somatotopy, while the others do not. In other words, the present results may be specific for a particular subgroup of amputees; this should be discussed.*

We agree that this question warrants further discussion. Unfortunately, the methods used in this study critically rely on the ability to move individual phantom digits in order to trigger the phantom hand representation. We therefore only recruited amputees with strong kinaesthetic sensation of each of the phantom digits. It would have been impossible to run the current study with individuals experiencing no (or partial) phantom hand sensations, as this would violate some of the assumptions underlying the travelling wave analysis. However, we can draw from our previously published findings conducted in a standard field (3 tesla) scanner (using a procedure that allowed us to scan individuals with varying degrees of phantom sensations (Makin et al., 2013; Makin et al., 2015). We showed that 16 of 17 individuals were able to activate the missing hand territory when moving (or attempting to move) their phantom hand. Although these studies were not suited to resolve whether this activity reflected topographic organisation, they hint that persistent representation may be the rule, rather than the exception. Nevertheless, it is still possible that the features of the preserved topography (e.g. overlap across digits, digit preferences) may depend upon (or drive) phantom sensations. In the revised manuscript we highlight our previous finding and discuss the potential causal link between phantom sensations and preserved topography. We are currently running a new study to tackle this important research question and hope that we will be able to shed more light on this interesting question in the future.

Revised text in the Discussion of the Main text:

“Our findings are based on the unique phenomenology of phantom sensations, and as such all tested amputees reported experiencing exceptionally vivid phantom sensations, allowing them to voluntarily move each of their phantom fingers. An open question remains whether a relationship exists between the experience of phantom sensations and preserved missing hand topography. Our previous findings, showing that phantom hand movements activate the missing hand territory in individuals experiencing varying levels of phantom sensation vividness (Makin et al., 2013; Makin et al., 2015) might indicate that this is a general phenomenon. However, we note that these previous studies do not provide information on whether the topographic features underlying the phantom-evoked activity were preserved. Further research is needed to determine whether the preservation of missing hand topography depends on (or gives rise to) the experience of phantom sensations, or whether our finding reflects a fundamental organising principle of the brain that is independent of experience.”

5) How well did the patients manage to move individual digits, and how can this be assessed, given that this movement was imaginary?

First, we would like to clarify the nature of phantom hand movements, with respect to motor imagery. It has previously been shown that phantom movements differ from imagined movement (including imagined movement of the phantom) both in terms of peripheral signals (i.e. electromyography, (Raffin, Giraux and Reilly, 2012)) and central signals (Raffin et al., 2012). In our study we instructed participants to move their phantom digits and ensured that our participants correctly interpreted these instructions (see also below). We believe this dissociation between motor execution and motor imagery to be essential for the interpretation of our findings and therefore clarify this important point in the revised manuscript.

Revised text in the Main text:

“Importantly, phantom movements are distinguishable from imagined movements. This is supported by empirical evidence demonstrating that phantom limb movements elicit both central and peripheral motor signals, that are different from those found during imagined movements (Makin et al., 2013; Reilly et al., 2006; Raffin et all, 2012).”

To ensure the amputees understood the task and were able to perform it well, we asked them to demonstrate (outside the scanner) the extent of volitional movement afforded with each of their phantom digits, by mirroring the phantom movements onto their intact hand. In addition, for the below elbow amputee, the experimenter palpated the stump muscles outside the scanner to verify that actual phantom movements were executed during the task. This procedure is described in detail in the Materials and methods, MRI tasks section. In the revised manuscript we also clarify this in the main text.

Revised text in the Main text:

“To ensure adequate task performance, amputees were asked to demonstrate to the experimenter outside the scanner the extent of volitional movement carried out in each of their phantom digits during the task, by mirroring the phantom movements with their intact hand.”

We also assessed the amputees vividness and difficulty to perform movements with the individual digits of the phantom hand using subjective ratings (Figure 1—figure supplement 2). In addition to this, we asked the amputees to describe the quality and extent of movement afforded in each of the phantom digits. In the revised manuscript, this information has been added to Figure 1—figure supplement 2.

*6) The complexity of the possible neurophysiological processes that underlie the fMRI results requires more consideration in an extended discussion, specifically with regard to the possible pathways that might have generated the brain activity observed here. The following three comments may be useful in this regard:*

We would like to thank the reviewers for these interesting, thought provoking, thoughtful and thorough comments. We greatly enjoyed corresponding with them, and believe that the revisions made based on these comments have greatly improved our Discussion.

*a) Equally as interesting as the preservation of the deafferented glabrous finger map in area 3b and 1, per se, is fact that these maps could be selectively locally activated, presumably by higher level postcentral somatosensory areas and/or by precentral motor areas. Understanding the pathway of activation is further challenged by the differences in the 3 cases presented here. In the below-elbow case, it is likely that portions of the extensor digitorum communis (finger raising) and flexor digitorum profundus (finger curling) muscles as well as their afferent and efferent innervation remain and were activated during the 'imagined' movements, which could probably be better described as 'executed' (muscle) movements. In this case, one might imagine the involvement of M-I to S-I connections, along with all the complexities of the difference between muscles located in the arm versus glabrous skin on the underside of the fingers. But in the above-elbow case, those finger muscles in the arm are gone; although dorsal root ganglion cells formerly innervating glabrous finger skin as well as motorneurons formerly innervating those muscles are likely to have persisted above the injury, and may have made ectopic connections. Finally, the brachial plexus avulsion case would likely have involved damage to the dorsal root ganglia and the spinal cord itself. In that case, the most peripheral source of abnormal neural activity could only have been neurons in the dorsal horn itself.*

The reviewers’ conveyed train of thought is entirely compatible with ours, and we are pleased to include this in the manuscript. In the revised manuscript we included a paragraph explaining the three cases presented in the manuscript and the potential mechanism driving the observed brain activity, given the differences between the three amputation cases.

Revised text in the Discussion of the Main text:

“Which inputs could contribute to the maintenance of the missing hand topography? The variability in the level and nature of amputations in this study’s cohort allows us to consider the potential contribution of the peripheral nervous system in the preservation of missing hand topography. In the below elbow amputee, some forearm muscles normally controlling hand movements are spared and therefore proprioceptive inputs relating to phantom hand movements likely persist (Nystrom and Hagbarth, 1981). In the above elbow amputee, these inputs would be absent, though ectopic firing from the injured nerve (Nystrom and Hagbarty, 1981) or intact dorsal root ganglia could preserve some afferent inputs (as previously shown following peripheral nerve injury (Nordin et al., 1984), see also (Vaso et al., 2014) for related findings). However, in the amputee suffering from brachial plexus avulsion injury, the dorsal root ganglia are damaged, meaning that no such peripheral input should be available. Given the observation of preserved topography in all three cases, it is highly probable that the preserved missing hand maps are not maintained by peripheral input, but rather are driven by processing in the central nervous system itself.”

*b) An important piece of information to keep in mind when interpreting these results is that there are several levels of map before we reach the cortex: dorsal root ganglion -> dorsal column nuclei -> ventrobasal nucleus -> S-I, and dorsal root ganglion -> dorsal column nuclei -> ventrolateral nucleus -> M-I cortex. Recent experiments on non-human primates with chronic lesions of the dorsal column subserving the arm have suggested that our initial expectations for sites of plastiticy may not have been correct (for example, P Chand and N Jain (2015) Intracortical and thalamocortical connections of the hand and face representations in somatosensory area 3b of macaque monkeys and effects of chronic spinal cord injuries. Journal of Neuroscience 35:13475-13486.) That study showed that despite the electrophysiological presence of large-scale somatotopic reorganization in 3b, surprisingly, there was no evidence for any marked increase in cortico-cortical connections across the hand-face border in 3b,* or *any evidence of altered map connections from the hand and face parts of the ventrobasal nucleus to the cortex. So the large-scale expansion of the chin representation into the deafferented hand representation in 3b may actually instead have been driven by the reorganization of the dorsal column nuclei. In the context of the present paper, this suggests that reorganized inputs to cortical maps should strongly be distinguished from the intrinsic organization of the cortical maps themselves, which contrary to expectations, may turn out to be* less *plastic than more peripheral stations."*

We agree that the manuscript could benefit from a mechanistic interpretation, and that the latest evidence for brainstem plasticity potentially provides a compelling framework to resolve our own findings with previous work on reorganisation. Specifically, our results are compatible with the notion that reorganisation may occur at subcortical areas, leaving the inherent organisation of SI relatively intact. We have added a paragraph in the Discussion of the revised manuscript explaining how to put together previous reorganisation studies and our result of preserved functional organisation in SI.

For revised text in the Discussion of the Main text, see response to reviewers’ point 3.

*c) Another point perhaps worth a sentence in the Discussion is that the experiment performed with the below-the-elbow amputee – having them contract their finger muscles without generating an stimulation of the (missing) glabrous skin surface of the finger is not straightforwardly possible with an intact limb since any activation of muscles that move the fingers will generate some direct somatosensory stimulation, even with the hand held up in the air. However, it might be possible to test this in an intact limb by anesthetizing an intact hand with a wrist block to see if cortical region previously thought to be mainly driven by stimulation of the glabrous surface of the fingers can be directly driven by efference copy (or other) signals.*

To rephrase the reviewers’ comment, it remains to be determined what the neural mechanism driving activation of the preserved missing hand maps is. Key to understanding this is the difference between imagery and actual phantom movements, as highlighted before in point 5. Indeed, while the sensory input is profoundly impacted following amputation, motor signals persist. Our amputees therefore provide a unique model to examine representation in the brain, without any sensory inputs. Since we show digit topography in primary somatosensory cortex in the physical absence of a hand, we believe our finding demonstrates the potential involvement of efference copies in driving these maps. We believe that the missing hand topography elicited in our study is the consequence of corollary discharge: information about descending commands that the motor system provides to the sensory system. In the revised manuscript we elaborate further on the potential role of efference copies in driving the missing hand maps (see also point 13).

Revised text in the Discussion of the Main text:

“What neural signals may be triggering the brain activations subserving the missing hand maps? Given the relatively unimpaired motor system in amputees, it is possible that these representations are driven by motor (efferent) information. The motor system is thought to provide information about its descending commands to the sensory system, by means of efference copy. When efference signals reach the sensory areas, they evoke activity in those areas. The pattern of this corollary discharge could resemble that of the sensory feedback to be expected from the movement (London and Miller, 2013). While predictive signals are fundamental components for current theories of motor control (Franklin and Wolpert, 2011), surprisingly little empirical evidence exists to demonstrate efferent signals in the primates’ S1 hand representation independently of afferent processing (London and Miller, 2013). Our evidence for digit topography in S1 despite the physical absence of a hand suggests the involvement of non-afferent processing in S1. The persistence of efference signals from the motor system could contribute to the maintenance of preserved structure and function in SI despite afferent input loss.”

Applying a sensory nerve block to an intact limb would indeed allow further investigation regarding the role of efference signals in eliciting detailed representation in primary somatosensory cortex. In the revised manuscript we did not elaborate on this interesting issue, as it seems to exceed the focus of the paper.

*d) The fact that phantom limb sensations can arise almost immediately during brachial blocks (Gentili ME, Verton C, Kinirons B, Bonnet F. (2002) Clinical perception of phantom limb sensation in patients with brachial plexus block. Eur J Anaesthesiol 19:105-108 also suggests that some 'plasticity' is immediate.*

We thank the reviewers for pointing us to this reference (8). We read the paper with great interest, but unfortunately we have to conclude that the authors are misusing the term “phantom sensations”. In the paper, the authors define phantom sensation as the inability to correctly describe the position of a limb following axillary block. This definition is very different from the way phantom sensations are described by amputees. Amputees describe a phantom sensation as the vivid experience that their missing limb is still present, often coinciding with the ability to move the phantom limb. Crucial to this point, amputees are able to describe the exact position of the phantom limb. As such we would prefer not to speculate whether phantom sensations could be induced by nerve blocks.

In our lab we use pharmacological nerve block in healthy controls and we can confirm that the experience that is reported following pharmacological nerve block is different from reports of amputees’ phantom sensations. This is consistent with microneurography studies showing persistent activity in the amputated nerve in amputees (Nystrom and Hagbarth, 1981), which would presumably be abolished during nerve block. In the revised manuscript we did not elaborate on this interesting issue, as it seems to exceed the focus of the paper.

References:

1. Flor H, Elbert T, Knecht S, Wienbruch C, Pantev C, Birbaumer N, et al. Phantom-limb pain as a perceptual correlate of cortical reorganization following arm amputation. Nature. 1995 Jun 8;375(6531):482–4.

2. Feldman DE, Brecht M. Map plasticity in somatosensory cortex. Science. 2005 Nov 4;310(5749):810–5.

3. Margolis DJ, Lütcke H, Schulz K, Haiss F, Weber B, Kügler S, et al. Reorganization of cortical population activity imaged throughout long-term sensory deprivation. Nature Neuroscience. Nature Publishing Group; 2012 Oct 21;15(11):1539–46.

4. Ramachandran VS, Stewart M, Rogers-Ramachandran DC. Perceptual correlates of massive cortical reorganization. Neuroreport. 1992 Jun 30;3(7):583–6.

5. Ramachandran VS. Behavioral and magnetoencephalographic correlates of plasticity in the adult human brain. Proc Natl Acad Sci USA. 1993 Nov 15;90(22):10413–20.

6. Knecht S, Henningsen H, Elbert T, Flor H, Höhling C, Pantev C, et al. Reorganizational and perceptional changes after amputation. Brain. 1996 Aug;119 (Pt 4):1213–9.

7. Grüsser SM, Mühlnickel W, Schaefer M, Villringer K, Christmann C, Koeppe C, et al. Remote activation of referred phantom sensation and cortical reorganization in human upper extremity amputees. Exp Brain Res. 2004 Jan;154(1):97–102.

8. Gentili ME, Verton C, Kinirons B, Bonnet F. Clinical perception of phantom limb sensation in patients with brachial plexus block. EJA. 2006 Aug 16;19(02):105.